# Unravelling the history of hepatitis B virus genotypes A and D infection using a full-genome phylogenetic and phylogeographic approach

Evangelia-Georgia Kostaki[1], Timokratis Karamitros[1,2], Garyfallia Stefanou[1], Ioannis Mamais[3], Konstantinos Angelis[1], Angelos Hatzakis[1], Anna Kramvis[4], Dimitrios Paraskevis[1]*

[1]Department of Hygiene, Epidemiology and Medical Statistics, Medical School, National and Kapodistrian University of Athens, Athens, Greece; [2]Department of Zoology, University of Oxford, Oxford, United Kingdom; [3]Department of Health Sciences, School of Sciences, European University of Cyprus, Nicosia, Cyprus; [4]Hepatitis Virus Diversity Research Unit, Department of Internal Medicine, Faculty of Health Science, University of the Witwatersrand, Johannesburg, South Africa

**Abstract** Hepatitis B virus (HBV) infection constitutes a global public health problem. In order to establish how HBV was disseminated across different geographic regions, we estimated the levels of regional clustering for genotypes D and A. We used 916 HBV-D and 493 HBV-A full-length sequences to reconstruct their global phylogeny. Phylogeographic analysis was conducted by the reconstruction of ancestral states using the criterion of parsimony. The putative origin of genotype D was in North Africa/Middle East. HBV-D sequences form low levels of regional clustering for the Middle East and Southern Europe. In contrast, HBV-A sequences form two major clusters, the first including sequences mostly from sub-Saharan Africa, and the second including sequences mostly from Western and Central Europe. Conclusion: We observed considerable differences in the global dissemination patterns of HBV-D and HBV-A and different levels of monophyletic clustering in relation to the regions of prevalence of each genotype.
DOI: https://doi.org/10.7554/eLife.36709.001

*For correspondence:
dparask@med.uoa.gr

**Competing interests:** The authors declare that no competing interests exist.

## Introduction

Hepatitis B virus (HBV) is the main cause of liver disease with an estimated number of 257 million people being chronically infected worldwide (*Schweitzer et al., 2015*). HBV belongs to the family *Hepadnaviridae*. The HBV genome consists of a partially double stranded DNA molecule approximately 3.2 kb long that replicates via an RNA intermediate. As a result of the activity of the error prone reverse transcriptase, HBV is characterized by a high degree of genetic heterogeneity and is classified into 9 genotypes (A-I) (*Kramvis, 2014*) and a putative 10th genotype J (*Tatematsu et al., 2009*), with an intergroup divergence of at least 7.5% across the complete genome. All genotypes, except for genotypes E and G, are further classified into subgenotypes with a mean genetic divergence of about 4% (*Schaefer, 2007*). These genotypes and subgenotypes can display complex ethnical and geographical distributions (*Kramvis, 2014*). Although genotypes D and A are omnipresent around the globe, genotype A prevails in Europe and Africa, with genotype D prevailing in Europe and the Middle East (*Schaefer, 2007*). Genotypes B and C are mostly found in Eastern Asia and Oceania, genotype E in Central and Western Africa, while genotypes F and H in Latin America and Alaska (*Norder et al., 2004*). Genotype D is considered to be pandemic (*Kramvis, 2014*).

**eLife digest** The Hepatitis B virus (HBV) is a major cause of liver disease, and according to the World Health Organization, around 257 million people live with Hepatitis B infection. The virus is a relatively ancient one in human history and has been infecting humans for at least 28,000 years. Previous studies have isolated HBV DNA from human skeletons dating from 800 to 7,000 years ago in Europe and Central Asia. Multiple types of this virus exist. Two types called HBV-A and HBV-D are present worldwide, with HBV-A being prevalent in Africa and in Europe, and HBV-D being very common in the Middle East and also in Europe.

Even though HBV has been infecting humans for millennia, there is little detailed knowledge of the how the disease spread among populations and geographical areas in the past. Due to few studies in this discipline, understanding of how the different types of HBV were dispersed and disseminated over time has remained patchy.

Now, Kostaki et al. analysed HBV-A and HBV-D DNA sequence data from present-day Hepatitis B patients to piece together a global map of historic spread of the virus. The results showed that HBV-D originated in North Africa and the Middle East, while HBV-A originated close to Africa and Europe and probably in the Middle East and Central Asia. HBV-A initially spread in Central Africa, after which it split into two separate pathways. The first spread to Sub-Saharan/eastern and southern Africa, with the other stretching to Sub-Saharan/eastern Africa. Much later, major regional transmissions happened from Africa to Brazil, Haiti and the Indian subcontinent, which are thought to be most likely due to the slave trade.

Uncovering the history of the spread of HBV and the human activities associated with it can help to inform public health strategies for avoiding similar situations happening again. These findings could be specifically useful in prevention of HBV in geographical areas where transmission is a high risk, ultimately helping to take steps toward eliminating HBV.

DOI: https://doi.org/10.7554/eLife.36709.002

Phylogenetic analysis of genotype D showed separation into nine distinct clusters [subgenotypes D1, D2, D3, Recombino-subgenotype (RS)-D4, RS-D5, RS-D6, RS-D7, RS-D8 and RS-D9 (formerly 'D4', 'D5', 'D6', 'D7', 'D8' and 'D9')] with high bootstrap support (*Pourkarim et al., 2010*).

Subgenotype D1 is dominant in North Africa, Europe, Western Asia, Indonesia and Australia (*Kramvis and Paraskevis, 2013*; *Bozdayi et al., 2005*; *Garmiri et al., 2011*; *Yousif and Kramvis, 2013*), D2 in the United Kingdom, Albania, Northeastern Europe, Russia, Malaysia and Japan (*Yousif and Kramvis, 2013*; *Tallo et al., 2008*; *Zehender et al., 2012*) and D3 predominates in Serbia, South Africa and the United States of America (USA) (*Yousif and Kramvis, 2013*; *De Maddalena et al., 2007*; *Lazarevic et al., 2007*). RS-D4 (formerly 'D4') is found in Haiti, the Arctic and Oceania (*Yousif and Kramvis, 2013*; *Norder et al., 2004*), RS-D5 (formerly 'D5') circulates mainly in indigenous populations in India (*Banerjee et al., 2006*), RS-D6 (formerly 'D6') in Tunisia, Morocco and Madagascar (*Kitab et al., 2011*; *Meldal et al., 2009*). RS-D8 and RS-D9 (formerly 'D8' and 'D9') have been found in Niger (*Abdou Chekaraou et al., 2010*) and India (*Ghosh et al., 2013*), respectively. An updated classification of genotype D has been published (*Pourkarim et al., 2010*; *Yousif and Kramvis, 2013*; *Pourkarim et al., 2014*).

Genotype A is predominant in Northwestern Europe, North America and sub-Saharan Africa. Although seven HBV-A subgenotypes (A1–A7) have been described in the literature, a comprehensive analysis of this genotype has resulted in an updated classification (*Pourkarim et al., 2010*). Genotype A is thus classified into subgenotypes A1, A2, A4, and Quasi-subgenotype (QS)-A3, because the latter group of sequences does not meet the criteria for a subgenotype classification (*Pourkarim et al., 2014*). A1 can be found in sub-Saharan Africa (South Africa, Congo, Tanzania, Malawi, Kenya, Zimbabwe, Uganda, Somalia), and in areas outside Africa where there was historically forced migration as a result of the slave trade (*Kramvis and Paraskevis, 2013*) including South Asia (India, Philippines, Bangladesh, Nepal) and South America (*Kramvis, 2014*; *Banerjee et al., 2006*; *Bowyer et al., 1997*; *Sugauchi et al., 2004*). A2 is mostly found in Europe and North America (*Norder et al., 2004*; *Bowyer et al., 1997*). QS-A3 (formerly 'A3', 'A4', 'A5') is frequently found in Gabon (*Makuwa et al., 2006*) and Cameroon ('A3') (*Kurbanov et al., 2005*), in Mali ('A4'), and in

Nigeria (*Olinger et al., 2006*) and Haiti ('A5') (*Andernach et al., 2009*). A4 (formerly 'A6') has been found in African-Belgian patients (*Pourkarim et al., 2010*) and QS-A3 (formerly 'A7') is circulating in Rwanda and Cameroon (*Hübschen et al., 2011*).

The clinical manifestation of HBV infection differs between individuals infected with genotype A compared to those infected with genotype D because of different molecular characteristics of these genotypes, especially in the precore region of the HBV genome. HBeAg-negative G1896A mutant chronic hepatitis B predominates in areas where genotype D prevails (*Hadziyannis, 2011*) because 1858T is positively associated with genotype D and negatively associated with genotype A, which has 1858C (*Kramvis et al., 2008*) and thus genotype A cannot develop precore G1896A. However, subgenotype A1 but not A2, has alternative mechanisms, which result in HBeAg-negativity (*Kramvis, 2016*). Consequently, subgenotype A2 shows the highest frequency of HBe-Ag-positivity compared to A1 and genotype D and this is highly statistically significant in individuals younger than 29 years (*Tanaka et al., 2004*). This means that subgenotype A2 has a longer high replicative, low inflammatory phase compared to subgenotype A1 and genotype D. A limited number of studies have shown that subgenotype A1, and perhaps genotype D are associated with an increased risk of developing serious complications of HBV in comparison with subgenotype A2 (*McMahon, 2009*; *Kew et al., 2005*; *Gopalakrishnan et al., 2013*), which is characterized by chronic HBV infection and sexual transmission (*Hadziyannis, 2011*; *Araujo et al., 2011*). Multivariate logistic regression analysis revealed only genotype A was independently associated with viral persistence following acute hepatitis B (*Japanese AHB Study Group et al., 2014*). Patients infected with genotype A respond better to interferon-based therapy compared to patients infected with genotype D (*Lin and Kao, 2010*; *Kramvis and Kew, 2005*). Although overall no significant difference in response of the different genotypes/subgenotypes to nucleos(t)ide analogue therapy has been found (*Lin and Kao, 2010*), response to adefovir may be lower in patients infected with subgenotype A2 because of the presence of L217R polymorphism in the S region (*Bottecchia et al., 2008*).

The epidemiological history of the HBV-D and HBV-A genotypes remains unclear because of the scarcity of relevant studies. In this study, we estimated the levels of regional clustering for HBV-D and HBV-A, in order to shed light on how these genotypes have been disseminated among geographic regions and countries over time. We also estimated their putative geographic origin and major dispersal pathways over the course of genotype A and D infection.

**Table 1.** Sampling of HBV[*] genotype D sequences from different geographic regions.

| Region of sampling | Countries (N) | Sequences (N, %) |
|---|---|---|
| North Africa/Middle East | 5 | 394 (43.0) |
| South Asia | 2 | 151 (16.5) |
| Western Europe | 9 | 70 (7.6) |
| Central Asia | 4 | 44 (4.8) |
| Australasia | 2 | 40 (4.4) |
| Asia-Pacific | 1 | 36 (3.9) |
| East Asia | 2 | 35 (3.8) |
| Eastern Europe | 3 | 27 (3.0) |
| Latin America | 2 | 25 (2.7) |
| Oceania | 6 | 23 (2.5) |
| Sub-Saharan Africa | 8 | 20 (2.2) |
| North America | 2 | 16 (1.8) |
| Central Europe | 2 | 14 (1.5) |
| Caribbean | 2 | 11 (1.2) |
| Southeast Asia | 2 | 10 (1.1) |
| Total | 52 | 916 (100) |

[*] HBV, *hepatitis B virus.*
DOI: https://doi.org/10.7554/eLife.36709.003

**Table 2.** HBV* genotype D percentages of clustering based on the geographic region of sampling.

| Region of sampling | Monophyletic clusters (N) | Clustered sequences (N, %) |
|---|---|---|
| Australasia | 1 | 36 (90) |
| Oceania | 1 | 19 (83) |
| Asia-Pacific | 1 | 30 (83) |
| Southeast Asia | 1 | 8 (80) |
| East Asia | 2 | 24 (69) |
| Caribbean | 1 | 7 (64) |
| Latin America | 1 | 14 (56) |
| Sub-Saharan Africa | 2 | 11 (55) |
| North America | 1 | 8 (50) |
| Western Europe | 3 | 21 (30) |
| South Asia | 3 | 34 (23) |
| Eastern Europe | 1 | 5 (19) |
| North Africa/Middle East | 6 | 63 (16) |
| Central Asia | 0 | 0 (0) |
| Central Europe | 0 | 0 (0) |
| Total | 24 | 279 (31) |

* HBV, *hepatitis B virus.*

DOI: https://doi.org/10.7554/eLife.36709.008

## Results

### HBV genotype D

To analyze the global dispersal patterns of HBV-D, we used 916 full-length, non-recombinant, unique sequences per patient (*Table 1*), because recombination analysis revealed evidence for mosaicism for 73 sequences out of the original dataset of 999 sequences. We also identified 10 multiple sequences, which were excluded from the analysis.

Conducting an all-embracing phylogenetic analysis using globally sampled, non-recombinant sequences, we showed that HBV genotype D formed regional clusters at different levels depending on the geographic origin of sampling (*Figure 1*, *Figure 1—figure supplement 1*). In detail, 90% of the Australasian sequences form a single monophyletic subcluster (D1), as is the case for sequences from Oceania (83%; RS-D4), Asia-Pacific (83%; D2), Southeast Asia (80%; RS-D6), Caribbean (64%; RS-D4), Latin America (56%; RS-D4), and North America (50%; D3) (*Table 2*). Sequences from East Asia (69%; D1 and D2) and sub-Saharan Africa (55%; D3 and RS-D8) formed two monophyletic subclusters each, while sequences from Western Europe (30%), South Asia (23%), Eastern Europe (19%), and North Africa/Middle East (16%) revealed the lowest monophyly levels (*Table 2*). The sequences from Central Europe and Central Asia did not present any monophyletic pattern (*Table 2*).

Country-wise analysis showed the following monophyly patterns: Greenland: 100% (D2), New Zealand: 97% (D1), Japan: 83% (D2), Tunisia: 66% (RS-D7) and China: 65% (D1) (*Supplementary file 1*). On the other hand, we observed very low monophyly patterns for Iran, Syria, Turkey, Belgium, India, Lebanon, and Russia (mostly sampled from the Asian part of Russia) (*Supplementary file 1*, *Figure 1—figure supplement 1*). The number of local transmission networks-LTNs (monophyletic clusters) for each country is shown in *Supplementary file 1*.

Phylogeographic analysis showed North Africa/Middle East as the putative origin of genotype D (*Figures 1* and *2*), however from the present dataset, the exact origin cannot be accurately inferred. Similarly, inclusion of RS-D5, for which discordant phylogenetic clustering was found (see Materials and Methods), did not provide stronger evidence about the origin of genotype D (*Figure 1—figure supplement 2*, *Figure 2—figure supplement 1*, *Figure 1—figure supplement 3*,). Major dispersal pathways for genotype D were complex including different geographic regions (*Figure 2*). Moreover, major clusters were connected by short internal nodes, suggesting similar genetic distances to

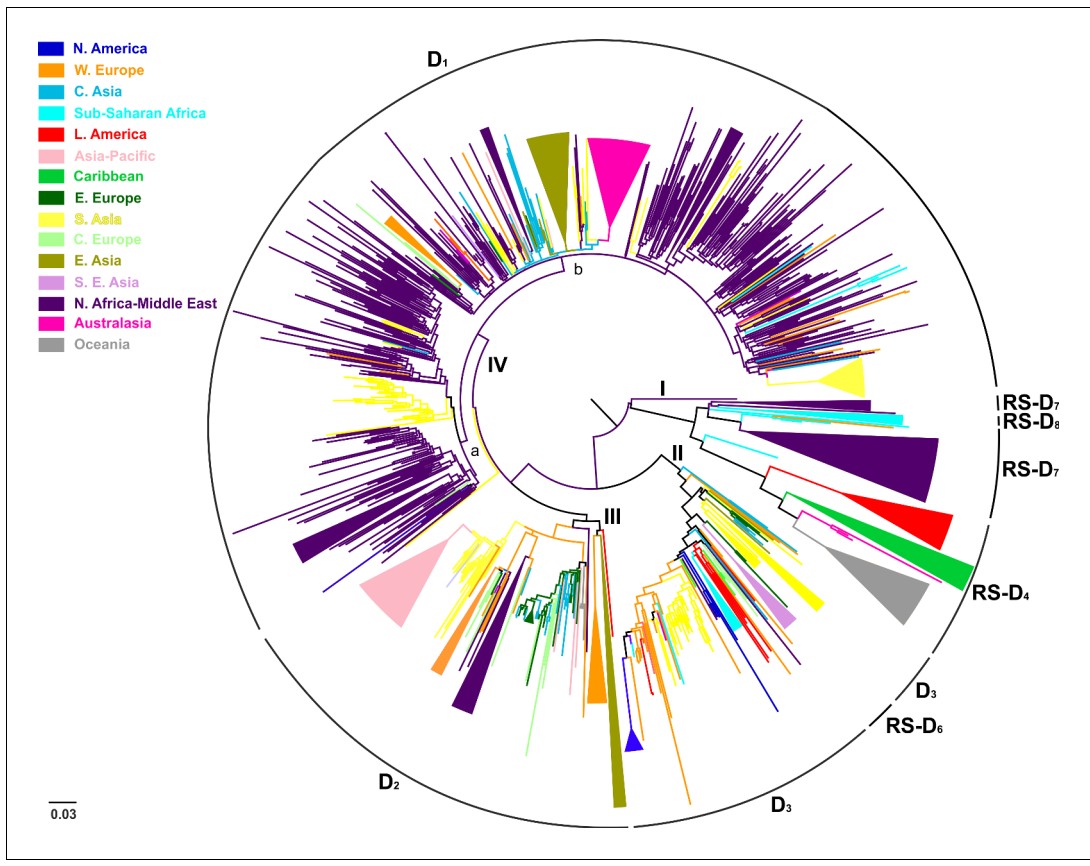

**Figure 1.** Midpoint rooted phylogeographic tree estimated by RAxML v8.0.20. HBV genotype D sequences (N = 916) categorized according to the geographic region of sampling were used in the analysis. Monophyletic clusters are indicated as triangles. Major clusters are indicated in Roman numerals.

DOI: https://doi.org/10.7554/eLife.36709.004

The following figure supplements are available for figure 1:

**Figure supplement 1.** Midpoint rooted phylogeographic tree on 916 HBV genotype D sequences categorized according to the country/geographic region of sampling.

DOI: https://doi.org/10.7554/eLife.36709.005

**Figure supplement 2.** Midpoint rooted phylogeographic tree on HBV genotype D sequences after the inclusion of RS-D5.

DOI: https://doi.org/10.7554/eLife.36709.006

**Figure supplement 3.** Phylogenetic analysis of genotype D including the RS-D5 in two non-overlapping subgenomic areas (1–2000 nts) and (2001–3078 nts). RS-D5 sequences are shown in red.

DOI: https://doi.org/10.7554/eLife.36709.007

the root of genotype D (*Figure 1*, *Figure 1—figure supplement 1*). Initial disseminations include the two subclusters (Tunisia, sub-Saharan Africa and Latin America/Caribbean, Australasia/Oceania) within the major clade I (RS-D7, RS-D8 and RS-D4) (*Figure 2*, *Figure 1—figure supplement 1*). This clade is distinct from the rest of the phylogeny that can be further divided into three major clusters (*Figures 1* and *2*, *Figure 1—figure supplement 1*). Clade II consisting mostly of viral sequences from South and Southeast Asia, North America (D3 and RS-D6), clade III including East Asia, Greenland, India, Russia and a large monophyletic clade from Japan (D2), and clade IV that can be further divided into subclade IVa and IVb (D1). Iran, Turkey, Syria and India dominate within subclade IVa, while in IVb, three large regional subclusters were found for China, New Zealand and India (*Figure 2*, *Figure 1—figure supplement 1*). Inclusion of RS-D5 results in an additional early pathway towards India (*Figure 1—figure supplement 2*, *Figure 2—figure supplement 1*, *Figure 1—figure supplement 3*).

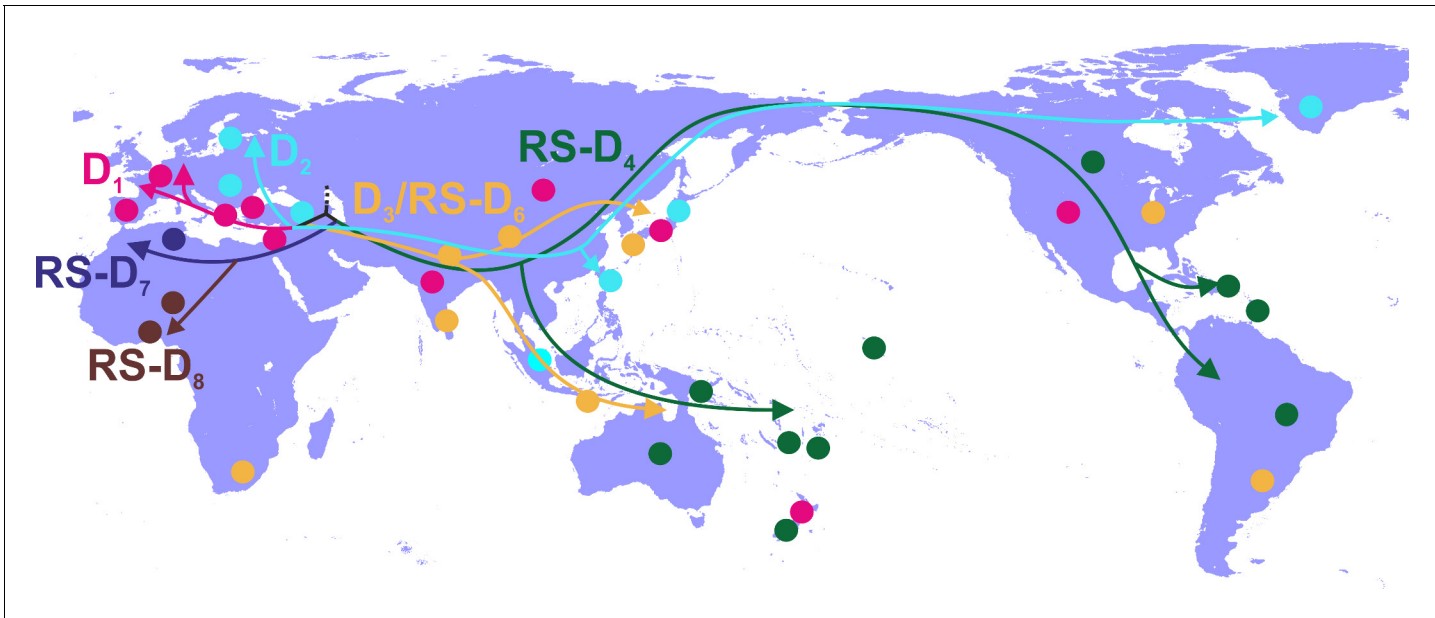

**Figure 2.** Putative major dispersal pathways of genotype D as estimated by phylogeographic analysis. The origin is depicted with dotted line. Subgenotypes and their corresponding dispersal routes are shown with different colors. Colored circles depict the geographic areas where subgenotypes are the most prevalent.

DOI: https://doi.org/10.7554/eLife.36709.010

The following figure supplement is available for figure 2:

**Figure supplement 1.** Putative major dispersal pathways of genotype D as estimated by phylogeographic analysis after the inclusion of RS-D5.

DOI: https://doi.org/10.7554/eLife.36709.011

## HBV genotype A

Of the 744 full-length genotype A sequences, evidence for inter-genotypic recombination and multiple sequences was found for 13 and 238 sequences, respectively, thus the final analysis included 493 sequences (*Table 3*). HBV genotype A formed two major clusters including sequences mostly from

**Table 3.** Sampling of HBV[*] genotype A sequences from different geographic regions.

| Region of sampling | Countries (N) | Sequences (N, %) |
|---|---|---|
| Sub-Saharan Africa | 14 | 112 (22.7) |
| Western Europe | 8 | 108 (21.9) |
| Latin America | 5 | 62 (12.6) |
| Asia-Pacific | 1 | 53 (10.8) |
| Caribbean | 2 | 52 (10.5) |
| Central Europe | 2 | 37 (7.5) |
| South Asia | 3 | 26 (5.3) |
| Eastern Europe | 4 | 17 (3.5) |
| North America | 2 | 11 (2.2) |
| Southeast Asia | 2 | 8 (1.6) |
| East Asia | 2 | 4 (0.8) |
| North Africa/Middle East | 2 | 2 (0.4) |
| Central Asia | 1 | 1 (0.2) |
| Total | 48 | 493 (100) |

[*] HBV, *hepatitis B virus.*

DOI: https://doi.org/10.7554/eLife.36709.012

**Table 4.** HBV* genotype A percentages of clustering based on the geographic region of sampling.

| Region of sampling | Monophyletic clusters (N) | Clustered sequences (N, %) |
|---|---|---|
| South Asia | 1 | 23 (88) |
| Caribbean | 3 | 44 (85) |
| Latin America | 4 | 44 (71) |
| Sub-Saharan Africa | 6 | 70 (63) |
| Western Europe | 1 | 33 (31) |
| Asia-Pacific | 2 | 11 (21) |
| Central Europe | 0 | 0 (0) |
| Eastern Europe | 0 | 0 (0) |
| Southeast Asia | 0 | 0 (0) |
| East Asia | 0 | 0 (0) |
| North America | 0 | 0 (0) |
| North Africa/Middle East | 0 | 0 (0) |
| Central Asia | 0 | 0 (0) |
| Total | 17 | 255 (52) |

* HBV, *hepatitis B virus.*

DOI: https://doi.org/10.7554/eLife.36709.013

sub-Saharan Africa (clade I; QS-A3 and A1), and Western Europe (clade II; A2) (*Figure 3*). Clade I consisted of three subclades, the first (subclade Ia; QS-A3) and the second (subclade Ib; A1) including sequences mostly from sub-Saharan Africa, while the third (subclade Ic; A1) from Caribbean, Latin America and South Asia (*Figure 3*). The highest levels of regional clustering were found for South Asia (88%; A1), Caribbean (85%, A1 and QS-A3), Latin America (71%; A1 and A2) and sub-Saharan Africa (63%; A1 and QS-A3) (*Table 4*). Sequences from Western Europe (31%) and Asia-Pacific (21%) showed low levels of regional dispersal (*Table 4*), whereas for Central and Eastern Europe, Southeast Asia, East Asia, North America, North Africa/Middle East and Central Asia no monophyletic clustering was detected (*Table 4*).

In a more detailed analysis (country-wise), we found two regional clusters (monophyletic subclusters/LTNs) within subclade Ia, which consist of sequences sampled from Haiti (N = 20; 1 LTN; QS-A3) and the Cameroon (N = 8; 1 LTN; QS-A3). Three regional clusters were found within subclade Ib (A1), consisting of sequences from South Africa (N = 43; 3 LTNs) (*Supplementary file 2*, *Figure 3—figure supplement 1*). Similarly, with regard to subclade Ic (A1), sequences from Haiti (N = 24; 2 LTNs) and Brazil (N = 20; 2 LTNs) formed four monophyletic clusters (*Supplementary file 2*, *Figure 3—figure supplement 1*). For clade II (A2), we found five regional clusters included samples from Japan (N = 11; 2 LTNs), Argentina (N = 6; 1 LTN), Belgium (N = 33; 1 LTN), and Panama (N = 13; 1 LTN) (*Supplementary file 2*, *Figure 3—figure supplement 1*). In addition, for some countries the patterns of monophyly were negligible (*Supplementary file 2*). The tree reconstruction revealed that sequences sampled from sub-Saharan Africa were located close to the root (*Figure 3*, *Figure 3—figure supplement 1*). With respect to the putative origin of genotype A, clade I was very close to the root of the tree, while the internal branch connecting clade II to the root was much longer than that for clade I (*Figure 3*, *Figure 3—figure supplement 1*). These findings suggest that viral diversity within clade I originated much earlier than clade II and closer to the date of the most recent common ancestor of genotype A (*Figure 4*). Phylogeographic estimations showed that the origin of clades I and II was in Africa (sub-Saharan Africa with the most probable areas in the Cameroon or neighboring countries) and in Europe (most probable areas France and Belgium), respectively (*Figures 3,4*, *Figure 3—figure supplement 1*).

With regard to the dispersal patterns of genotype A over the course of time, at the early stages, it followed two distinct pathways within Africa (clade I) and the one that later gave rise to the European infection (clade II) (*Figure 4*). The separate phylogenetic branching of clades I and II and the rest of genotype A sequences together with the fact that they have spread into two different geographic areas suggest that the putative origin of genotype A is close to Africa and Europe and

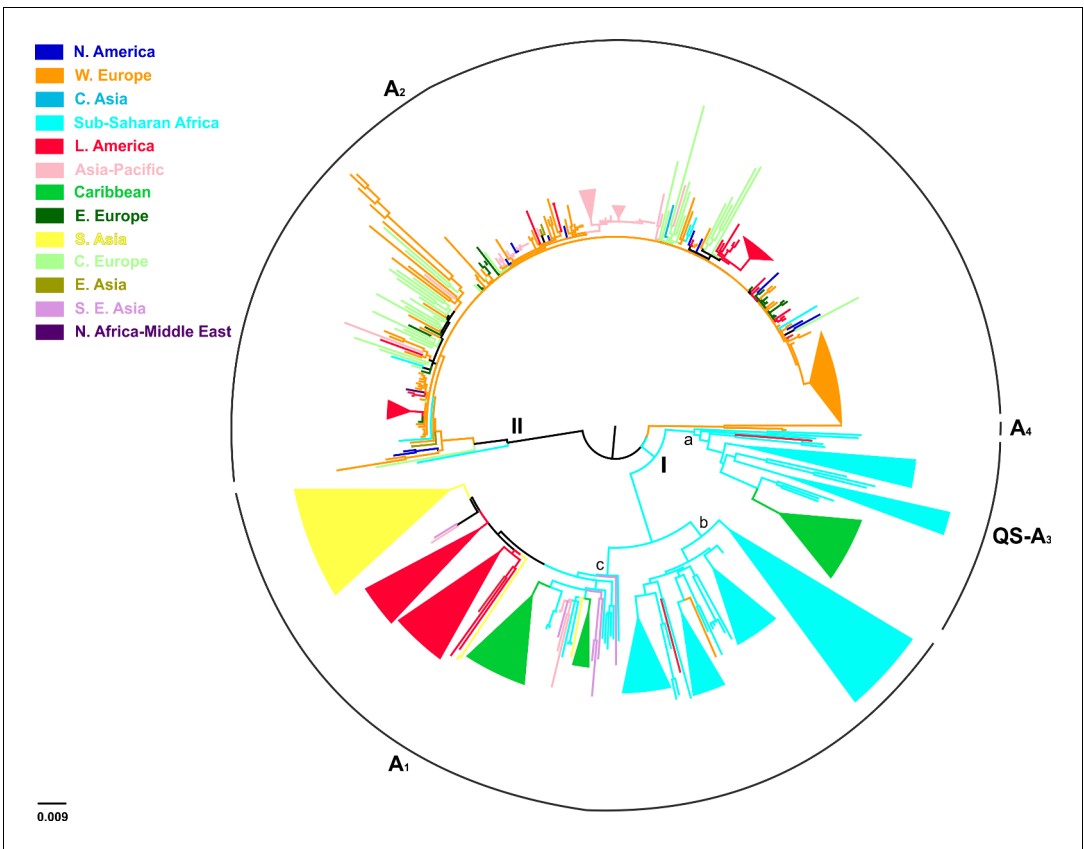

**Figure 3.** Midpoint rooted phylogeographic tree estimated by RAxML v8.0.20. HBV genotype A sequences (N = 493) categorized according to the geographic region of sampling were used in the analysis. Monophyletic clusters are indicated as triangles. Major clusters are indicated in Roman numerals.

DOI: https://doi.org/10.7554/eLife.36709.014

The following figure supplement is available for figure 3:

**Figure supplement 1.** Midpoint rooted phylogeographic tree on 493 HBV genotype A sequences categorized according to the country/geographic region of sampling.

DOI: https://doi.org/10.7554/eLife.36709.015

probably in the Middle East/Central Asia (*Figure 4*). After the initial spread in Africa (Central Africa) (subclade Ia), the virus followed two distinct pathways: one to sub-Saharan Africa/Eastern Africa and southern Africa, (subclade Ib) and another to sub-Saharan Africa/Eastern Africa (subclade Ic) (*Figure 4*). Spillovers lead to major regional transmissions, which occurred within subclade Ia to Haiti; and within subclade Ic towards Brazil, Haiti and South Asia (India and other countries in South Asia) (*Figure 4*). The infection in Europe originated as a result of a distinct pathway for which the intermediate events are unknown. Further spread occured to different parts within Europe as well as to areas further afield in America such as the USA, Argentina and Panama and in a few cases in Japan (*Figure 4*).

## Discussion

We used the maximum likelihood method with bootstrap evaluation to reconstruct the phylogeny of globally sampled, full-length, non-recombinant sequences of HBV genotypes D and A. The description of the spatial characteristics of viral dispersal plays a pivotal role in understanding the history of the respective infections and to make hypotheses about the parameters potentially associated with the observed dispersal patterns. We also performed phylogeographic analysis to estimate the origin of HBV lineages and the pathways of dispersal over the course of the genotype A and D infections.

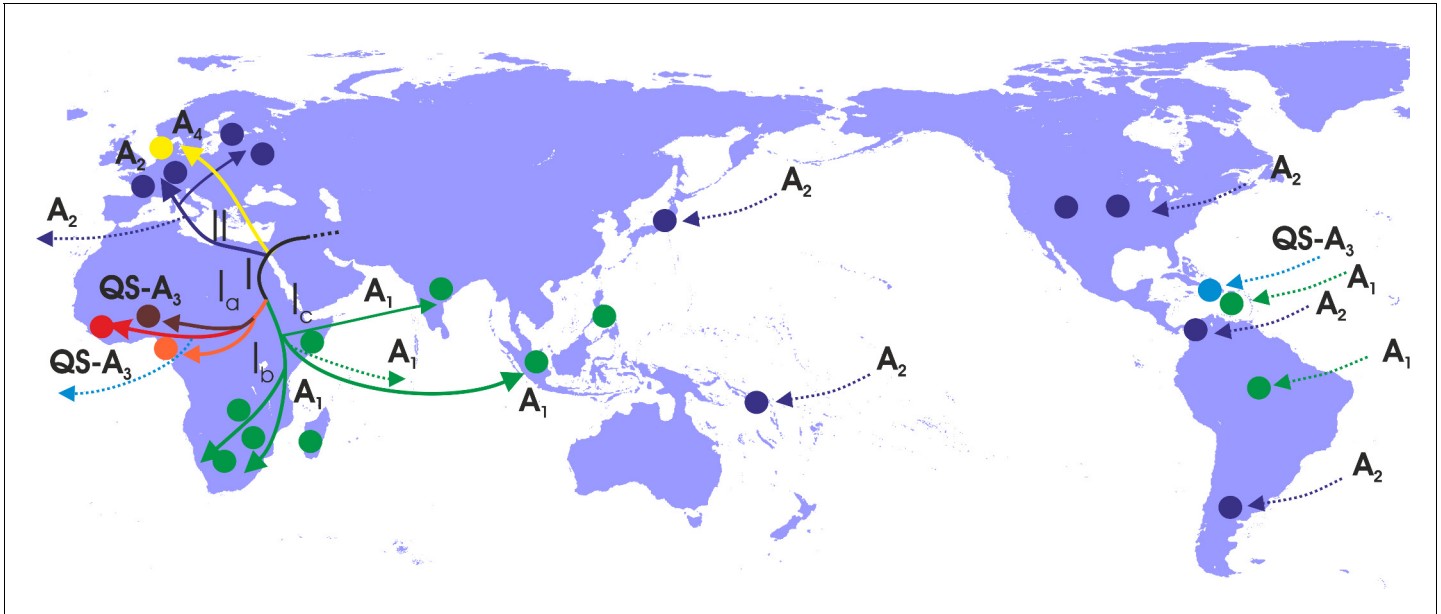

**Figure 4.** Putative major dispersal pathways for genotype A as revealed by the phylogeographic analysis. The putative geographic origin of genotype A is in the Middle East/Central Asia. Subgenotypes and their corresponding dispersal routes are shown with different colors. Colored circles depict the geographic areas where subgenotypes are the most prevalent. Dotted lines represent the source and sinks for distant dispersal pathways.
DOI: https://doi.org/10.7554/eLife.36709.016

Previous studies supported the 'Out of Africa' model for the origin and expansion of HBV by providing evidence of HBV/Humans co-expansion and co-development of their population sizes 22,000–47,100 years ago (*Paraskevis et al., 2013*), while estimating the origin of HBV in humans at 34,100 (27,600–41,300) years ago (*Paraskevis et al., 2015*). The ancient origin of HBV has been also confirmed by analysis of HBV sequences from two 16th century mummies and two recent studies, which detected HBV from ancient DNA samples, ranging in age from approximately 800 to 4,500 years (*Mühlemann et al., 2018*) and 1,000 to 7,000 years ago (*Krause-Kyora et al., 2018*), showed that the HBV infections were present for at least 7,000 years (*Patterson Ross et al., 2018*; *Kahila Bar-Gal et al., 2012*). As the major genotypes have probably originated before and during the onset of Neolithic and the subgenotypes during the later Neolithic period, the majority of HBV diversity has been accumulated as a result of dispersals following the antecedent 'Out of Africa' population migrations, which hosted and conveyed the parental HBV strains (*Kramvis, 2014*; *Locarnini et al., 2013*; *Zehender et al., 2014*).

It is important to note that rapid human population expansion occurred outside of Africa, following a radial pattern across the Eurasian continents. More specifically, Central Asia served as a node for the dispersal of Humans towards North Africa, the Eastern Mediterranean and South Europe and Southeast Asia before reaching the Americas (*Henn et al., 2012*). HBV follows the same routes, with serial founder events creating the major genotypes, which in turn become prevalent in a compartmental pattern of dispersal around the globe (*Paraskevis et al., 2013*, *2015*). However, it is expected that regional monophyletic clustering abundance would not be identical and would be dependent on the levels of host mobility. For example, regional monophyletic clustering is found in geographic areas, which remained isolated for a long time (e.g. subgenotype B6 in the Arctic, subgenotypes C3, C4, RS-D4 in Oceania and the Pacific) (*Kowalec et al., 2013*; *Osiowy et al., 2011*; *Lusida et al., 2008*; *Thedja et al., 2011*) suggesting that the HBV dispersal was the result of onward transmissions of a single or few strains introduced in the population(s) in the past (*Paraskevis et al., 2013*). High proportions of strains not belonging to monophyletic clusters suggest significant population mobility giving rise to infections; whereas dominant regional monophyletic clustering suggests that an epidemic has been introduced from a single or few restricted sources (as a result of limited human and viral mobility).

In the present study, considerable differences regarding the patterns of dispersal and the regions of clustering of HBV genotypes D and A around the globe were identified. For HBV-D, we found low levels of regional clustering for North Africa/Middle East, suggesting high levels of viral mobility. This is in line with our previous observations, about the co-expansion of HBV with its host and the central role of North Africa and the Middle East regions as hubs for human expansion and consequent dissemination and genetic shuffling of genotype D (*Paraskevis et al., 2013*). Notably we also found that this area provides the putative origin for genotype D. This conclusion is supported by the finding that the location of the HBV sequence that clustered at the root of genotype D and sampled 2,300 years ago, was in Central Asia (data not shown) (*Mühlemann et al., 2018*). On the other hand, regarding Greenland and New Zealand, we found almost 100% monophyletic clustering suggesting that the genotype D infection in these areas were because of onward transmissions among the local population(s) and not due to introduction from recent human migrations into these areas. The high levels of monophyly clustering in East Asia (Japan, China) and North Africa (Tunisia) suggest a limited number of genotype D introductions in these areas. The high viral genetic diversity for these areas supports the hypothesis that the founder strains were not introduced because of recent human migrations following globalization during the 20th century. In contrast, India showed a low level of regional clustering suggesting a highly mobile infection. Moreover, the phylogeographic trees suggest that virus has been disseminated across different populations, probably as a result of extensive human mobility, at different time periods.

Our analysis revealed that HBV-A shows a strong pattern of regional dispersal with both macro- or micro-levels of clustering. Macro-level clustering showed that genotype A can be further grouped into two major clusters (African and European). These clusters correspond to subgenotype A1 in Africa and subgenotype A2 outside of Africa (*Kramvis and Paraskevis, 2013*). The putative origin of genotype A was probably in the Middle East/Central Asia and thereafter followed two distinct routes of dispersal, one within Africa and a second to Western Europe. The hypothesis that the putative origin of genotype A was in the Middle East/Central Asia was further supported by the recent analysis of HBV ancestral genotype A sequences in ancient samples (*Mühlemann et al., 2018*). Specifically, the location of two HBV sequences, which clustered at the root of the genotype A and sampled 4,000 years ago, was in Central Asia (data not shown) (*Mühlemann et al., 2018*). Supporting the results of our previous study, where we analysed subgenomic preS/S region, HBV-A strains from sub-Saharan Africa were the source for Caribbean, Latin America and South Asia following recent human mobility, as a result of the historical forced migration of the slave trade in the 16th – 19th centuries (*Kramvis and Paraskevis, 2013*). Similarly, from group II different regional clusters were detected for Western Europe and Latin America. Notably, the HBV-A in the USA had a Western European origin. This contrasts with Caribbean and some areas in Latin America, where HBV originated in Africa. Regional dispersal was dominant in Panama, Cameroon, South Africa and Rwanda.

The African dispersal pathways had, probably, a sub-Saharan origin from Central Africa and thereafter moving southwards. Introduction of HBV in Europe was the result of a founder effect occurring later than in Africa. The origin of parental European strains is missing. Notably genotype A has been introduced to Haiti, Brazil and the Indian subcontinent as a result of slave trade (*Kramvis and Paraskevis, 2013*).

In the current study, we analyzed the dispersal patterns of two of the most globally disseminated HBV genotypes. To our knowledge, this is one of the few studies showing the dispersal pathways based on the phylogeographic analysis of all full-length available data for genotypes A and D. The highest levels of clustering for genotype A suggest limited viral mobility at the earliest phase of this infection. Thereafter regional transmissions remained dominant as supported by high levels of monophyly. In contrast to HBV-A, for the exception of Tunisia, limited regional dispersal was found for genotype D in North Africa/Middle East, while this region acted as its putative source. Limited monophyly for these areas was observed at micro-level clustering as well suggesting high levels of mobility over the course of the infection. These findings can be explained by the fact that the area acted as a source for HBV-D but also the areas of the earliest dispersal showed high levels of human mobility for a number of reasons including the expansion of agriculture during the Neolithic revolution, the development of modern civilizations and the existence of major trade routes (*Diamond and Bellwood, 2003*). In contrast, during the same period, human mobility was relatively limited in sub-Saharan Africa, thus leading to regional dispersal of HBV-A. Our findings are corroborated by the recently published analysis of HBV from ancient samples (*Mühlemann et al., 2018*; *Krause-*

*Kyora et al., 2018*), which show that HBV was already circulating in humans at least 7,000 years ago and, in agreement with our analyses, place the putative origin of genotypes A and D in the Middle East/Central Asia.

Our study has several limitations mostly related to potential sampling bias of HBV sequences available in the databases and the lack of information about country of birth or the immigration status of patients for whom HBV sequences were included in the current study. However, we do not expect that these limitations affect the levels of monophyly per country and the proposed dispersal pathways of the genotypes A and D.

In conclusion, the observed differences of the dispersal patterns and the levels of regional clustering between HBV-D and HBV-A around the globe, probably portray the impact of the prehistoric human activities on the evolution of this pathogen, but also highlight the importance of co-evolution of the host in phylogenetic reconstructions of slowly evolving pathogens such as HBV (*Paraskevis et al., 2015*).

## Materials and methods

### DNA sequences, alignment and HBV genotyping

We downloaded all available full-length HBV sequences found in public repositories National Center for Biotechnology Information (NCBI; http://www.ncbi.nlm.nih.gov) and the Hepatitis B virus database (HBVdb; https://hbvdb.ibcp.fr/HBVdb/) for genotype D (*N* = 999) and genotype A (*N* = 733) with available geographic area of sampling. Data were collected between May 2014 and December 2015. Duplicate sequences from NCBI and HBVdb were removed if they had the same accession number. Information about geographic area of sampling for each sequence was retrieved from the NCBI database. Detailed information about the country of birth or the immigration status of patients, from whom HBV sequences were included in the analysis, was not available. Alignment for each genotype was performed by MUSCLE as implemented in MEGA v7 (*Kumar et al., 2016*).

We checked all published papers for the presence of an 'outbreak'. This was reported for three studies of genotype D, one from Germany (N = 1 sequence) (*Petzold et al., 1999*), one from the USA (N = 7 sequences) (*Garfein et al., 2004*) and one from India (N = 39) (*Arankalle et al., 2011*) (*Supplementary file 3*). The outbreak sequences corresponded to 5.1% (47 of 916) of the genotype D sequences. Similarly, for genotype A 'outbreaks' were reported in two studies, one from the USA (N = 3 sequences) (*Parekh et al., 2003*) and one from Belgium (N = 58 sequences) (*Pourkarim et al., 2009*) (*Supplementary file 4*), corresponding to 12.3% (61 of 493) of the genotype A sequences analyzed in our study.

We also looked for the presence of multiple sequences from individual patients and we found that for two cases: (i) (N = 7 clones; country of sampling: India) and (ii) (N = 5 clones; country of sampling: Italy), multiple sequences were available for genotype D (unpublished data). We also found that for two studies: (i) [N = 238 multiple sequences from four patients; country of sampling USA (*Thai et al., 2012*)] and (ii) [N = 5 multiple sequences; country of sampling Germany (*Hass et al., 2005*)], multiple sequences were included for genotype A. We kept only a sequence per patient in the analysis.

HBV genotypes were confirmed by the Oxford HBV Automated Subtyping Tool v1.0 (*Alcantara et al., 2009*) and phylogenetic analysis using as references 110 sequences from all previously known HBV genotypes downloaded from the NCBI database. Phylogenetic analysis was performed using the approximate maximum likelihood method with the Generalized Time Reversible (GTR + cat) model of nucleotide substitution model including a gamma (Γ) distributed rate of heterogeneity among sites parted on 20 categories as implemented in FastTree v2.1 program (*Price et al., 2010*).

### HBV nomenclature

In order to be consistent with the updated HBV nomenclature (*Pourkarim et al., 2010*, *2014*), we present our results using the updated nomenclature system. According to the new system, formerly introduced subgenotypes 'A3', 'A4' and 'A5' have been named as QS-A3 (*Pourkarim et al., 2010*). The reasons for the new classification were: (i) full-length genomic regions were unavailable; (ii) inter-subgenotypic nucleotide divergence was <4%, and (iii) monophyletic clustering was supported by

weak bootstrap values (*Pourkarim et al., 2014*). The previously introduced 'A6' was named 'A4 '(*Pourkarim et al., 2014*). With regard to genotype D, previously named 'D4', 'D5', 'D6', 'D7', 'D8', 'D9' were classified as RS-D4, RS-D5, RS-D6, RS-D7, RS-D8, RS-D9, respectively. This was due to the putative recombinant nature of these subgenotypes (*Yousif and Kramvis, 2013*; *Meldal et al., 2009*; *Ghosh et al., 2013*, *2012*).

Our analysis, as explained in the next paragraph, showed that subgenotypes RS-D4, RS-D6, RS-D7 and RS-D8 were not found to be recombinants. However, to avoid confusion for the readers we decided not to change the existing HBV nomenclature.

## Recombination analysis and country grouping

We used specialized programs (Oxford HBV Automated Subtyping Tool v1.0, RDP4 v4.36, Simplot v3.5.1) in order to detect the recombinant forms of the virus (*Martin et al., 2015*; *Lole et al., 1999*). In order, to analyze the global dispersal patterns of HBV-D and HBV-A the final datasets consisted of 916 and 493 full-length, non-recombinant and non-redundant sequences, respectively. Recombination analysis detected sequences of subgenotype RS-D9 to be recombinants (*Ghosh et al., 2013*). On the other hand, sequences of subgenotypes RS-D4, RS-D6, RS-D7 and RS-D8 were not found to be recombinants and therefore they have been included in our analyses. Recombination analysis included bootscanning plots for each subgenotype against genotypes A, B, C, D, F, G and H. Genotype E was not included in the analysis because of its recombinant nature with D (*Simmonds and Midgley, 2005*). Analysis was performed in two steps including: (i) only pure subgenotypes D1, D2 and D3 and (ii) all subgenotypes of genotype D expect the query. Recombination analysis was also performed using RDP4 v4.36, which combines many diverse recombination detection methods and is one of the most sensitive tools to detect recombination, against the same dataset of references. Notably, we found no evidence of recombination using bootscanning and RDP4 analysis for RS-D4, RS-D6, RS-D7 and RS-D8. The potential reason for their previous classification as recombinants (i.e. RS-D7, RS-D8) was that they were found to consist of a partial genotype E region, however, in this region, all genotype D sequences cluster with genotype E (data not shown). Therefore, this phylogenetic relationship with genotype E is not unique to the previously reported recombinant subgenotypes but occurs for all genotype D sequences. Therefore, subgenotypes RS-D4, RS-D6, RS-D7 and RS-D8 were included in our analysis since there was no evidence for recombination. Recombination analysis using bootscanning and RDP4 for RS-D5 showed that several RS-D5 sequences were intragenotypic recombinants consisting of diverse mosaic patterns. Phylogenetic analysis of the two subgenomes of the full-length HBV genome (1–2000 and 2001–3078 nts; corresponding to sites 60–2059, 2060–3179 of the reference X02496) revealed discordant phylogenetic signal with regard to the clustering of RS-D5. Specifically, in the first part of the genome (1–2000 nts) RS-D5 clustered as an outlier to genotype D, while in the second half of the genome (2001–3078 nts) RS-D5 clustered within genotype D. The discordant clustering of the RS-D5 prompted us to perform phylogenetic analysis with and without RS-D5.

## Phylogenetic and phylogeographic analysis

Phylogeny reconstruction with bootstrap evaluation was conducted by the maximum likelihood method for each genotype separately, using the GTR + G nucleotide substitution model as implemented in RAxML v8.0.20 (*Stamatakis, 2014*). We defined as monophyletic clusters as those having bootstrap values higher than 70%, within which 70% of strains share the same geographic area (country or a geographic region) of sampling. Trees were converted to midpoint rooted by using the FigTree v1.4.2 program (http://tree.bio.ed.ac.uk/software/figtree/). The origin of genotypes A and D was inferred by character reconstruction using parsimony on the estimated maximum likelihood phylogeny using Mesquite v3.2 (*Maddison and Maddison, 2017*). We conducted two kinds of phylogeographic analyses; one grouping sequences according to country of sampling and another, grouping them according to large geographic areas as defined by the Global Burden of Disease classification system (http://www.who.int). The geographic areas are described in *Supplementary files 5* and *6* for genotypes D and A, respectively. The dispersal pathways were estimated as follows: Mesquite reconstructs the geographic origin of different clades (viral lineages), which in combination with their hierarchical clustering, provide evidence about the putative dispersal pathways of the virus. For example, for genotype A there are two major branches (I and II) for which

their geographic origin was estimated in Africa and Western Europe, respectively. The finding about two geographically distinct lineages point to the fact that the early dispersal occurred through two different pathways in the corresponding regions. The putative pathways can be further unraveled in a similar way as soon as we move downstream from the root to the tips.

## Acknowlegments

The study was in part funded by the Hellenic Scientific Society for the Study of AIDS and Sexually Transmitted Diseases

## Additional information

### Funding

| Funder | Author |
| --- | --- |
| Hellenic Scientific Society for the Study of AIDS and STDs | Angelos Hatzakis |

The funders had no role in study design, data collection and interpretation, or the decision to submit the work for publication.

### Author contributions

Evangelia-Georgia Kostaki, Timokratis Karamitros, Formal analysis, Investigation, Writing—original draft; Garyfallia Stefanou, Konstantinos Angelis, Data curation, Formal analysis, Investigation, Writing—review and editing; Ioannis Mamais, Data curation, Formal analysis, Methodology, Writing—review and editing; Angelos Hatzakis, Funding acquisition, Investigation, Writing—review and editing; Anna Kramvis, Formal analysis, Investigation, Writing—original draft, Writing—review and editing; Dimitrios Paraskevis, Conceptualization, Formal analysis, Supervision, Funding acquisition, Investigation, Writing—original draft

### Author ORCIDs

Evangelia-Georgia Kostaki (iD) http://orcid.org/0000-0002-3346-0930
Timokratis Karamitros (iD) http://orcid.org/0000-0003-0841-9159
Dimitrios Paraskevis (iD) http://orcid.org/0000-0001-6167-7152

### Decision letter and Author response

Decision letter https://doi.org/10.7554/eLife.36709.026
Author response https://doi.org/10.7554/eLife.36709.027

## Additional files

### Supplementary files

• Supplementary file 1. Sampling and percentages of clustering of HBV genotype D sequences from different countries and geographic regions.
DOI: https://doi.org/10.7554/eLife.36709.009

• Supplementary file 2. Sampling and percentages of clustering of HBV genotype A sequences from different countries and geographic regions.
DOI: https://doi.org/10.7554/eLife.36709.017

• Supplementary file 3. List of papers for HBV genotype D sequences included in the analysis.
DOI: https://doi.org/10.7554/eLife.36709.018

• Supplementary file 4. List of papers for HBV genotype A sequences included in the analysis.
DOI: https://doi.org/10.7554/eLife.36709.019

• Supplementary file 5. List of countries within each geographic region (as defined by the Global Burden of Disease classification system), in which HBV genotype D sequences were included in the analysis.

DOI: https://doi.org/10.7554/eLife.36709.020

• Supplementary file 6. List of countries within each geographic region (as defined by the Global Burden of Disease classification system), in which HBV genotype A sequences were included in the analysis.

DOI: https://doi.org/10.7554/eLife.36709.021

• Transparent reporting form

DOI: https://doi.org/10.7554/eLife.36709.022

## Data availability

All data (sequence alignments and additional pieces of information related to the accession numbers of sequences and their sampling areas) are available at Dryad (doi: 10.5061/dryad.bt4q242).

The following dataset was generated:

| Author(s) | Year | Dataset title | Dataset URL | Database, license, and accessibility information |
|---|---|---|---|---|
| Kostaki E, Karamitros T, Stefanou G, Mamais I, Angelis K, Hatzakis A, Kramvis A, Paraskevis D | 2018 | Data from: Unravelling the history of hepatitis B virus genotypes A and D infection using a full-genome phylogenetic and phylogeographic approach | http://dx.doi.org/10.5061/dryad.bt4q242 | Available at Dryad Digital Repository under a CC0 Public Domain Dedication |

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
