## [Decision Letter]

Thank you for submitting your article "Unravelling the epidemic history of hepatitis B virus genotypes A and D using a full-genome phylogeographic approach" for consideration by *eLife*. Your article has been reviewed by three peer reviewers, including Carla Osiowy as the Reviewing Editor and Reviewer #1, and the evaluation has been overseen by Gisela Storz as the Senior Editor. The following individual involved in review of your submission has agreed to reveal their identity: Federico García (Reviewer #3).

The reviewers have discussed the reviews with one another and the Reviewing Editor has drafted this decision to help you prepare a revised submission.

Summary:

The manuscript describes an analysis of global phylogeny and regional clustering of HBV genotypes A and D based on full genome, non-recombinant sequences available and downloaded from public sequence databases. Using phylogenetic methods, they have examined the clustering for these sequences to identify regional differences, the putative geographic origin and global dissemination of HBV A and D subgenotypes. Genotype A and D putatively originated in sub-Saharan Africa (Abstract, and Discussion, second and fifth paragraphs) and the region of North Africa/Middle East, respectively, and dispersal resulted in regional prevalence of specific sub-genotypes observed today. The article is well written and very interesting, particularly within the field of HBV and viral evolution, but would hopefully be of interest to evolutionary biologists, archaeologists, etc., due to the unique origin and evolutionary features of HBV serving as a complementary anthropological tool. In general the conclusions are reasonable from the results presented. There are only a few comments or concerns regarding aspects that require correction and regarding the methodology and result interpretation.

Essential revisions:

1) Study results regarding regional clustering and putative genotype origin are based on the geographic location as provided by submissions to public sequence databases. How reliable is this information? In particular, how did you control for database submissions in which the researcher's country of residence (or location of research) was listed as opposed to the patient's region of birth or residence prior to immigration (if relevant)? Referring to the Materials and methods subsection “DNA sequences, alignment and HBV genotyping”, the authors have downloaded all NCBI sequences for Genotypes A and D, then analysed these with a reference set of 110 sequences downloaded from NCBI. Did they remove any duplicated sequences from these 2 datasets? The authors also need to explain how duplicated sequences were identified and removed from the dual datasets from NCBI and HBVdb.

2) Further to comment 1, there is some concern regarding potential sampling bias. Were sequences from heterochronous sampling included in the analysis? Multiple sequences from single patients may skew prevalence estimates and thus overestimate regional clustering and more important, assumptions of monophyly. Similarly, database submissions related to a local common source transmission/outbreak may also overestimate monophyletic regional clustering. Was validation of the database selection performed to control for this? Some discussion is warranted regarding the potential limitation and bias that may be introduced.

3) The phylogeographic analysis performed did not result in molecular clock or temporal properties of the sequences. Yet inference is made regarding dispersal among migrating populations over time in the Discussion. How was this reconciled with the data? The authors had provided a message to reviewers regarding a new article describing the detection and analysis of HBV in ancient human bone and teeth samples. Going forward, it will be important to refer or discuss these findings in light of the present study results. For example, Mühlemann et al. and another recent paper (Krause-Kyora et al., 2018, *eLife,* ahead of print May 10 "Neolithic and Medieval virus genomes reveal complex evolution of Hepatitis B") describe HBV genotypes existing in locations inconsistent with present day distribution. Similarly, extinct lineages suggest a level of diversity that is no longer observable when estimating past origin and dispersal patterns using modern samples. Thus these new insights from the recent articles could be discussed within the manuscript and interpretation of results clarified.

4) It would be preferable to use the most current, validated naming system for all HBV subgenotypes throughout the manuscript to support appropriate naming/classification and avoid confusion for the reader. This has the benefit of reducing the number of supplementary figures (as well, Figures 1A and 2A and Supplementary Figures 1A and 2A appear to be almost identical – the supplementary figures are not needed). For example, in the Introduction, the sub-genotype numbering for genotype D needs to be better explained with regard to the previous and current sub-genotype designations. This has been explained well for genotype A in the next paragraph where the "former" sub-genotypes are also indicated, but this needs to be included for the genotype D paragraph for clarity. In the Materials and methods subsection “HBV nomenclature”, the authors state that they present their results using both the earlier and updated nomenclature, but have not done this for genotype D sub-genotypes. It may be easier to comprehensively explain the earlier and updated nomenclature in the Introduction, or Materials and methods, and then use the updated nomenclature throughout the rest of the paper, included the figures.

5) Materials and methods subsection “Recombination analysis and country grouping”. The authors indicate their finding that sub-genotype D5 was recombinant. This has not been previously noted and requires explanation or discussion. Also, sub-genotype D8 was not found to be recombinant, although this sequence has previously been noted as a recombinant sequence. This also needs an explanation.

6) Figure 1B needs correction with regard to the sub-genotype D4 sequences. According to Figure 1A, the only D4 sequences are from Latin America, the Caribbean, Australasia and Oceania. However, the coloured circles in Figure 1B indicate that sub-genotype sequences for D4 are also found in South Asia, South East Asia, East Asia and North America. Where are the sequences to explain this figure?

7) Figure 2B indicates the putative origin of genotype A is Middle East/Central Asia, as shown similarly to Figure 1B for genotype D, however the comments in the text suggest the origin was in Sub-Saharan Africa, and the figure legend indicates the exact geographic origin is unknown. The figure should be modified to reflect this.

---

## [Author Response]

Essential revisions:1) Study results regarding regional clustering and putative genotype origin are based on the geographic location as provided by submissions to public sequence databases. How reliable is this information? In particular, how did you control for database submissions in which the researcher's country of residence (or location of research) was listed as opposed to the patient's region of birth or residence prior to immigration (if relevant)? Referring to the Materials and methods subsection “DNA sequences, alignment and HBV genotyping”, the authors have downloaded all NCBI sequences for Genotypes A and D, then analysed these with a reference set of 110 sequences downloaded from NCBI. Did they remove any duplicated sequences from these 2 datasets? The authors also need to explain how duplicated sequences were identified and removed from the dual datasets from NCBI and HBVdb.

We went through all published studies reporting HBV sequences used in the analysis (Supplementary files 3 and 4). Detailed information about the country of birth of the HBV infected patients was not available. The following sentence was included in the Materials and methods: “Detailed information about the country of birth or the immigration status of patients for whom HBV sequences included in the analysis was not available”. The lack of information about the country of birth or the immigration status of patients for whom HBV sequences was mentioned as a potential limitation of our study.

Duplicate sequences from NCBI and HBVdb were removed if they had the same accession number.

The previous sentence was included in the Materials and methods.

2) Further to comment 1, there is some concern regarding potential sampling bias. Were sequences from heterochronous sampling included in the analysis? Multiple sequences from single patients may skew prevalence estimates and thus overestimate regional clustering and more important, assumptions of monophyly. Similarly, database submissions related to a local common source transmission/outbreak may also overestimate monophyletic regional clustering. Was validation of the database selection performed to control for this? Some discussion is warranted regarding the potential limitation and bias that may be introduced.

We checked all published papers for the presence of an “outbreak”. This was reported for three studies of genotype D, one from Germany (N=1 sequence; PMID=10223539), one from the USA (N=7 sequences, PMID: 15382123) and one from India (N=39, PMID 21108697) (Supplementary file 3). The outbreak sequences corresponded to 5.1% (47 of 926) of the genotype D sequences. Similarly, for genotype A “outbreaks” were reported in two studies, one from USA (N=3 sequences, PMID: 12767980) and one from Belgium (N=58 sequences, PMID: 19615936) (Supplementary file 4), corresponding to 8.3% (61 of 731) of the genotype A sequences analysed in our study.

We also looked for the presence of multiple sequences from individual patients and we found: i) for two cases (N=7 clones; country of sampling: India) and (N=5 clones; country of sampling: Italy) multiple sequences were available for genotype D (data available on NCBI but not published in scientific journal). We also found for two studies: i) (N=238 multiple sequences from 4 patients; country of sampling USA; PMID: 22510694) and ii) (N=5 multiple sequences; country of sampling Germany; PMID: 15962285) that multiple sequences were available for genotype A.

The previous text was included in Materials and methods. We repeated the whole analysis for both genotypes and we revised the text and the figures.

3) The phylogeographic analysis performed did not result in molecular clock or temporal properties of the sequences. Yet inference is made regarding dispersal among migrating populations over time in the Discussion. How was this reconciled with the data? The authors had provided a message to reviewers regarding a new article describing the detection and analysis of HBV in ancient human bone and teeth samples. Going forward, it will be important to refer or discuss these findings in light of the present study results. For example, Mühlemann et al. and another recent paper (Krause-Kyora et al., 2018, eLife, ahead of print May 10 "Neolithic and Medieval virus genomes reveal complex evolution of Hepatitis B") describe HBV genotypes existing in locations inconsistent with present day distribution. Similarly, extinct lineages suggest a level of diversity that is no longer observable when estimating past origin and dispersal patterns using modern samples. Thus these new insights from the recent articles could be discussed within the manuscript and interpretation of results clarified.

We discussed the new findings about the analysis of HBV from ancient samples from both papers in the discussion. We also discuss the importance of these findings with regard to our findings and the inference of the putative origin of genotypes A and D.

4) It would be preferable to use the most current, validated naming system for all HBV subgenotypes throughout the manuscript to support appropriate naming/classification and avoid confusion for the reader. This has the benefit of reducing the number of supplementary figures (as well, Figures 1A and 2A and Supplementary Figures 1A and 2A appear to be almost identical – the supplementary figures are not needed). For example, in the Introduction, the sub-genotype numbering for genotype D needs to be better explained with regard to the previous and current sub-genotype designations. This has been explained well for genotype A in the next paragraph where the "former" sub-genotypes are also indicated, but this needs to be included for the genotype D paragraph for clarity. In the Materials and methods subsection “HBV nomenclature”, the authors state that they present their results using both the earlier and updated nomenclature, but have not done this for genotype D sub-genotypes. It may be easier to comprehensively explain the earlier and updated nomenclature in the Introduction, or Materials and methods, and then use the updated nomenclature throughout the rest of the paper, included the figures.

We revised the text accordingly and we introduced the most current, validated naming system for all HBV subgenotypes. The number of supplementary figures has been reduced. We included a section in “Phylogenetic and phylogeographic analysis” paragraph about the revised nomenclature and the extensive recombination analysis performed for all putative recombinant subgenotypes of D. We found that there is no evidence for recombination for D4, D6, D7 and D8 subgenotypes.

5) Materials and methods subsection “Recombination analysis and country grouping”. The authors indicate their finding that sub-genotype D5 was recombinant. This has not been previously noted and requires explanation or discussion. Also, sub-genotype D8 was not found to be recombinant, although this sequence has previously been noted as a recombinant sequence. This also needs an explanation.

Recombination analysis using bootscanning and RDP4 for RS-D5 showed that several RS-D5 sequences were intragenotypic recombinants consisting of diverse mosaic patterns. Phylogenetic analysis of the two subgenomes of the full-length HBV genome (1-2000 and 2001-3078 nts; corresponding to sites 60-2059, 2060-3179 of the reference X02496) revealed discordant phylogenetic signal with regard to the clustering of RS-D5. Specifically, in the first part of the genome (1-2000 nts) RS-D5 clustered as an outlier to genotype D, while in the second half of the genome (2001-3078 nts) RS-D5 clustered within genotype D. The discordant clustering of the RS-D5 prompted us to perform phylogenetic analysis with and without RS-D5. We also report the results of the phylogeographic analysis and the dispersal pathways of genotype D after the inclusion of RS-D5 in the results and in separate figures (Figure 1—figure supplement 2, Figure 1—figure supplement 3, Figure 2—figure supplement 1)

6) Figure 1B needs correction with regard to the sub-genotype D4 sequences. According to Figure 1A, the only D4 sequences are from Latin America, the Caribbean, Australasia and Oceania. However, the coloured circles in Figure 1B indicate that sub-genotype sequences for D4 are also found in South Asia, South East Asia, East Asia and North America. Where are the sequences to explain this figure?

Figure 1B was revised (it now appears as Figure 2). We analysed D4 sequences from Latin America (Brazil), the Caribbean (Martinique, Haiti), Australasia (Australia, New Zealand), Oceania (Tonga, Kiribati, Fiji, Papua New Guinea, Samoa) and North America (Canada). Sequences from North America (Canada) clustered within the green LTN from the Caribbean (Martinique, Haiti) presented in Figure 1.

7) Figure 2B indicates the putative origin of genotype A is Middle East/Central Asia, as shown similarly to Figure 1B for genotype D, however the comments in the text suggest the origin was in Sub-Saharan Africa, and the figure legend indicates the exact geographic origin is unknown. The figure should be modified to reflect this.

We revised the text accordingly and the figure legend (Figure 2B appears now as Figure 4). We introduced the following sentence in the results to explain why the putative origin of genotype A was in Middle East/Central Asia: **“**The separate phylogenetic branching of clades I and II and the rest of genotype A sequences together with the fact that they have spread into two different geographic areas suggest that the putative origin of genotype A is close to the Africa and Europe and probably in Middle East/Central Asia (Figure 4)”.